# Obesity Is Associated with the Severity of Periodontal Inflammation Due to a Specific Signature of Subgingival Microbiota

**DOI:** 10.3390/ijms242015123

**Published:** 2023-10-12

**Authors:** Sylvie Lê, Sara Laurencin-Dalicieux, Matthieu Minty, Justine Assoulant-Anduze, Alexia Vinel, Noor Yanat, Pascale Loubieres, Vincent Azalbert, Swann Diemer, Remy Burcelin, Thibault Canceill, Charlotte Thomas, Vincent Blasco-Baque

**Affiliations:** 1Département d’Odontologie, Faculté de Santé, Université Paul Sabatier Toulouse III, 3 Chemin des Maraîchers, CEDEX 9, 31062 Toulouse, France; sylvie.le@inserm.fr (S.L.); sara.laurencin-dalicieux@univ-tlse3.fr (S.L.-D.); matthieu.minty@inserm.fr (M.M.); alexia.vinel@inserm.fr (A.V.); nonoyanat@gmail.com (N.Y.); pascale.loubieres@inserm.fr (P.L.);; 2Service d’Odontologie Toulouse Rangueil, CHU Toulouse, 3 Chemin des Maraîchers, CEDEX 9, 31062 Toulouse, France; 3UMR1297 Inserm, Team InCOMM/Intestine ClinicOmics Metabolism & Microbiota, Institut des Maladies Métaboliques et Cardiovasculaires (I2MC), Université Paul Sabatier, 1 Avenue Jean Poulhes, 31432 Toulouse, France; assoulan@insa-toulouse.fr (J.A.-A.); vincent.azalbert@inserm.fr (V.A.); swann.diemer@inserm.fr (S.D.); remy.burcelin@inserm.fr (R.B.); 4INSERM U1295, CERPOP, Epidémiologie et Analyse en Santé Publique, Risques, Maladies Chroniques et Handicaps, 37 Allées Jules Guesdes, 31000 Toulouse, France; 5UMR 1297 Inserm, Team ESTER, Institut des Maladies Métaboliques et Cardiovasculaires (I2MC), Université Paul Sabatier, 1 Avenue Jean Poulhes, 31432 Toulouse, France

**Keywords:** oral microbiota, subgingival microbiota, periodontitis, obesity, periodontal inflammation, PISA, periodontal treatment response

## Abstract

The aim of this study was to analyze the link between periodontal microbiota and obesity in humans. We conducted a cohort study including 45 subjects with periodontitis divided into two groups: normo-weighted subjects with a body mass index (BMI) between 20 and 25 kg/m^2^ (*n* = 34) and obese subjects with a BMI > 30 kg/m^2^ (*n* = 11). Our results showed that obesity was associated with significantly more severe gingival inflammation according to Periodontal Inflamed Surface Area (PISA index). Periodontal microbiota taxonomic analysis showed that the obese (OB) subjects with periodontitis were characterized by a specific signature of subgingival microbiota with an increase in Gram-positive bacteria in periodontal pockets, associated with a decrease in microbiota diversity compared to that of normo-weighted subjects with periodontitis. Finally, periodontal treatment response was less effective in OB subjects with persisting periodontal inflammation, reflecting a still unstable periodontal condition and a risk of recurrence. To our knowledge, this study is the first exploring both salivary and subgingival microbiota of OB subjects. Considering that OB subjects are at higher periodontal risk, this could lead to more personalized preventive or therapeutic strategies for obese patients regarding periodontitis through the specific management of oral microbiota of obese patients.

## 1. Introduction

Obesity is recognized as a major public health problem; 39% of the world’s adult population is overweight or obese, according to the WHO in 2017. Its evolution is considered pandemic. Indeed, obesity and associated cardio-metabolic diseases (cardiovascular pathologies and type 2 diabetes, for example) have reached alarming levels and continue to rise. However, mechanisms of action have yet to be elucidated. Recent data have identified the intestinal microbiota as a key player, but the oral microbiota has received less attention in the context of obesity [1,2,3]. However, the oral cavity is the second largest reservoir of microbiota in the body. It has been the subject of intensive study for several years, with evidence of its involvement in general pathologies such as diabetes, cardiovascular disease and obesity [4]. Evidence has suggested a relationship between periodontitis and obesity. Periodontitis is a chronic, non-communicable inflammatory disease linked to oral dysbiosis. It causes irreversible destruction of tooth-supporting tissues, ultimately leading to tooth loss. The positive association between obesity and periodontal disease has been demonstrated in numerous clinical and epidemiological studies, as well as in meta-analyses [5,6,7]. Obese individuals have a higher risk of developing severe periodontitis, but individuals with periodontitis also tend to have a higher body mass index (BMI). In their meta-analysis, Chaffee et al. showed a higher prevalence of obesity in subjects with periodontal disease [7]. Moreover, in obese patients, if non-surgical periodontal therapy improves local periodontal clinical parameters, the response to treatment remains less significant than in normo-weighted patients [8,9,10]. However, periodontal therapy combined with dietary weight loss improves the response of obese subjects to periodontal treatment and reduces systemic inflammatory parameters (C-Reactive Protein, pro-inflammatory cytokines) [11,12]. Indeed, systemic pro-inflammatory cytokines tend to decrease after weight loss [13]. Several mechanisms have been proposed to explain the link between obesity and periodontitis [14]. The immuno-inflammatory process is one of them. The classification of periodontal and peri-implant diseases and conditions highlights the predominant role of obesity via periodontal and systemic inflammation [15,16]. Obesity, characterized by a chronic low-grade inflammatory state, leads to an increase in pro-inflammatory mediators in periodontal tissues. Obese subjects with periodontitis are characterized by an increase in pro-inflammatory cytokines (IL-1β, IL-6, TNF), also in blood, and also in gingival fluid samples, compared to normo-weighted subjects with periodontitis [17,18]. In the crevicular fluid of obese patients, we could detect an increase in pro-inflammatory adipokines (leptin, visfatin and resistin) [19]. The low-grade systemic inflammation associated with obesity increases periodontal inflammation and promotes bone resorption processes caused by the recruitment of immuno-inflammatory cells and certain cytokines (IL-1β, TNF) [20]. Some studies suggest that the dysbiotic microbiota present in periodontitis may play a role in obesity via the translocation of oral bacteria into the bloodstream, resulting in metabolic endotoxemia. This endotoxemia, defined as an elevation of plasma levels of lipopolysaccharides (LPS), major components of Gram-negative bacterial outer membranes, combined with a chronic immuno-inflammatory response, may contribute to the development and/or aggravation of metabolic disorders [4]. The molecular mechanism through which the microbiota controls energy metabolism is linked to the production of LPS in bloodstream. Our previous results showed a link between food intake and increased endotoxemia [21,22,23]. The metabolic endotoxemia by infusion of LPS targeted the adipose tissue by an increase in LPS and pro-inflammatory cytokines inside adipose cells. This modification of the local environment of adipose tissue promoted an infiltration of macrophages (F4/80-positive cells) and upgraded the stockage of lipids inside adipose cells [24]. Saito et al. proposed the bidirectional link between obesity and dysbiotic microbiota with Gram-negative bacteria and periodontitis [25]. We have recently shown that obese patients with periodontitis have a different salivary microbiota than that of normo-weighted patients with periodontitis [26]. The question remains as to whether they have an overall specific signature of their oral microbiotas. The principal aim of this study was to analyze the subgingival microbiota of obese subjects with periodontitis. For this matter, we conducted a cohort study of 45 patients with periodontitis divided into two groups: normo-weighted subjects with a body mass index (BMI) between 20 and 25 kg/m^2^ (*n* = 34) and obese subjects with a BMI > 30 kg/m^2^ (*n* = 11).

## 2. Results

### 2.1. Obese People with Periodontitis Have More Severe Periodontal Inflammation According to Periodontal Inflamed Surface Area (PISA) Index

The general parameters of the participants are presented in Table 1. The average age was similar in both normo-weighted (NW) and obese patients (OB) groups (47.56 years ± 10.62 vs. 53.64 years ± 11.96, respectively; *p* = 0.49). No significant differences between obese subjects and normo-weighted subjects were observed as far as the smoking status, stress score and oral hygiene habits were concerned.

To explore the link between obesity and oral health, different clinical parameters were analyzed (Table 1). The Decayed–Missing–Filled (DMF) index was similar between both groups (11.06 ± 5.35 vs. 11 ± 6.68; *p* > 0.99). Concerning periodontal parameters, clinical attachment loss (5.06 mm ± 2.05 vs. 4.64 mm ± 1.24; *p* = 0.40), probing depth (3.72 mm ± 1.16 vs. 3.74 mm ± 0.88; *p* = 0.95), bleeding on probing (33.91% ± 21.14 vs. 28.71% ± 15.05; *p* = 0.37) and plaque index (16.09% ± 13.87 vs. 12.12% ± 14.82; *p* = 0.44) were similar between NW and OB subjects. However, the Periodontal Inflamed Surface Area (PISA) index, which reflects the surface area of inflamed tissue found in hemorrhagic pockets, was significantly higher in OB subjects compared to NW ones (1677.83 mm^2^ ± 1489.92 vs. 986.99 mm^2^ ± 591.47; *p* = 0.049). To describe the link between the severity of the periodontal inflammation and obesity, we performed a Pearson’s correlation between PISA index and BMI. We observed a positive and significantly linear correlation between the PISA index and BMI (R^2^ = 0.06; *p* = 0.03) (Appendix A). We thus hypothesized that there is a specific signature of the subgingival microbiota in obese patients associated with a variation of PISA index.

### 2.2. Obese Patients Show a Specific Signature of Subgingival Microbiota Associated with a Decrease in Microbiota Diversity and an Increase in Periodontal Inflammation

To explore the interaction between clinical parameters and subgingival periodontal microbial parameters related to obesity, we performed a multivariate analysis by Principal Component Analysis (PCA) (Figure 1A). The results showed that the two main clinical factors responsible for the dispersion of OB and NW subjects according to the BMI in the PCA analysis were the PISA index and the Periodontal Simpson Index (alpha diversity index in periodontal microbiota) (Figure 1B). PCA and a Pearson’s correlation analysis reported that PISA index was negatively and significantly correlated with Periodontal Simpson Index (R^2^ = 0.10; *p* = 0.02) (Figure 1C). Furthermore, the beta diversity was significantly different between the periodontal microbiota of normo-weighted and those of obese patients (*p* = 0.012) (Figure 1D).

To evaluate the interaction between periodontal microbiota and obesity, we performed a taxonomic analysis of the periodontal microbiota in both groups: OB and NW (Figure 1E and Appendix A). The relative abundance of the Porphyromonadaceae (8.80% ± 10.84 vs. 17.44% ± 12.89; *p* = 0.034), Prevotellaceae (4.96% ± 8.58 vs. 12.28% ± 10.11; *p* = 0.0049), Tannerellaceae (3.64% ± 2.67 vs. 7.00% ± 5.74; *p* = 0.049) and Paludibacteraceae (0.72% ± 0.93 vs. 4.10% ± 5.09; *p* = 0.013) families was significantly lower in OB compared to NW patients. On the other hand, the relative abundance of the Streptococcaceae (8.13% ± 7.24 vs. 3.00% ± 4.52; *p* = 0.013) and Neisseriaceae (10.61% ± 19.01 vs. 1.38% ± 2.97; *p* = 0.022) families was significantly higher in OB compared to NW subjects (Figure 1E).

To determine the bacteria family in periodontal microbiota associated with the variation of the BMI, we performed a volcano plot analysis, presented in Figure 1F. Figure 1F and G represent a volcano plot analysis showing significant bacterial abundance according to the BMI and PISA indexes. We observed that the relative abundance of Gram-positive bacteria families such as Staphylococcaceae (*p* = 0.004), Lactobacillaceae (*p* = 0.005) and Micrococcaceae (*p* = 0.045), as well as Veillonellaceae (*p* = 0.027), was significantly increased in concert with BMI. Similarly, we determined the bacteria family in periodontal microbiota associated with the variation in periodontal inflammation (PISA index) (Figure 1G). We observed that the relative abundance of the Staphylococcaceae family (*p* = 0.003) was significantly increased in concert with the PISA index, whereas the relative abundance of the Flavobacteriaceae family (*p* = 0.043) was significantly increased according to a decrease in the PISA index. We also noted that Staphylococcaceae, a Gram-positive bacteria family, increased with the PISA index and that Flavobacteriaceae, a Gram-negative bacteria family, was more abundant with a decrease in the PISA index.

Obese subjects with periodontitis have a specific signature of their subgingival periodontal microbiota characterized by an increase in Gram-positive bacteria and a decrease in Gram-negative bacteria. Uncertainty remains about the origin of the presence of these Gram-positive bacteria. One hypothesis is that salivary Gram-positive bacteria, because of the local inflammation, could translocate into the periodontal pockets. We thus performed the taxonomic analyses of the salivary microbiota.

### 2.3. The Specific Signature of Periodontal Microbiota in Obese Subjects Is Close to the Composition of the Salivary Microbiota Independently of Obesity with an Increase in Gram-Positive Bacteria

To explore the interaction between periodontal and salivary microbiota in obesity, we performed a taxonomic analysis. We compared the relative abundance between both microbiotas within each group (OB subjects and NW subjects) (Appendix A).

We observed that the relative abundance of Neisseriaceae (18.16% ± 25.45 vs. 1.38% ± 2.97; *p* = 0.00019), Streptococcaceae (20.32% ± 13.90 vs. 3.00% ± 45.52; *p* < 0.0001) and Micrococcaceae (4.34% ± 10.29 vs. 0.78% ± 2.58, *p* > 0.0001) families in salivary microbiota of normo-weighted subjects (sNW) was significantly higher than in periodontal microbiota of normo-weighted subjects (pNW). In contrast, the relative abundance of Fusobacteriaceae (5.86% ± 5.24 vs. 17.99% ± 10.03; *p* < 0.0001), Porphyromonadaceae (5.46% ± 5.90 vs. 17.44% ± 12.89; *p* < 0.0001), Peptostreptococcaceae (0.84% ± 0.99 vs. 5.48% ± 6.01; *p* < 0.0001) and Tannelleraceae (0.31% ± 0.40 vs. 7.00% ± 5.74; *p* < 0.0001) families was significantly lower in sNW compared to pNW (Figure 2A,B). In obese subjects, we observed that the relative abundance of Prevotellaceae (11.22% ± 8.81 vs. 4.96% ± 8.58; *p* = 0.047), Neisseriaceae (31.52% ± 30.58 vs. 10.61% ± 19.01, *p* = 0.016), Streptococcaceae (18.49% ± 17.18 vs. 8.13% ± 7.24, *p* = 0.04), Pasteurellaceae (5.78% ± 7.21 vs. 3.17% ± 7.49; *p* = 0.01) and Micrococcaceae (3.76% ± 3.02 vs. 2.69% ± 3.74; *p* = 0.047) families in saliva microbiota of obese subjects (sOB) was significantly higher than in periodontal microbiota of obese subjects (pOB). Moreover, the relative abundance of Fusobacteriaceae (6.22% ± 7.59 vs. 15.73% ± 13.40; *p* = 0.034), Peptostreptococcaceae (0.50% ± 0.73 vs. 6.53% ± 7.54; *p* = 0.0032) and Tannelleraceae (0.20% ± 0.29 vs. 3.64% ± 2.67; *p* = 0.0004) families was significantly lower in sOB compared to pOB (Figure 2C,D). In addition, the overall distribution of Gram-positive and Gram-negative bacteria within each group was compared (Figure 2E). In normo-weighted subjects with periodontitis, the proportion of Gram-positive bacteria was significantly higher in the salivary microbiota than in the periodontal microbiota (32.94% vs. 12.21%; *p* = 0.0006). On the other hand, in obese subjects with periodontitis, a similar proportion of Gram-positive bacteria was observed in the salivary and periodontal microbiota (28.27% vs. 26.42%; *p* = 0.87). The periodontal microbiota of obese subjects had a higher proportion of Gram-positive bacteria than periodontal microbiota of normo-weight subjects (26.42% vs. 12.21%; *p* = 0.018). The relative abundance of Staphylococcaceae was significantly higher in periodontal microbiota of OB compared to periodontal microbiota of NW subjects (respectively, 1.13% ± 1.69 vs. 0.48% ± 1.99; *p* = 0.006) (Appendix A). The relative abundance of Staphylococcaceae was around 4.4-fold less important in the periodontal microbiota than in the salivary microbiota in normo-weight subject. In contrast, no change was observed between both microbiota in obese subjects.

When looking at the global distribution of the beta diversity of the four groups (sNW, pNW, sOB and pOB), three clusters stood out. This analysis showed a clear superposition between sNW and sOB microbiota, indicating no main differences in beta diversity between the two groups (*p* = 0.36). pNW microbiota represent a clear, unique cluster compared to sNW (*p* = 0.0005) and sOB (*p* = 0.0.001) microbiota. Interestingly, pOB microbiota was another unique cluster compared to other groups (pOB vs. pNW, *p* = 0.0005; pOB vs. sOB, *p* = 0.004; pOB vs. sNW, *p* = 0.005), confirming that the periodontal microbiota signature in obese subjects is close to that of the salivary microbiota (Figure 2F).

Where the composition of the periodontal microbiota of NW subjects is different from that of the salivary microbiota, in OB subjects, it seems to share similarities. This difference in microbiota could have an impact on periodontal treatment response.

### 2.4. Periodontal Treatment Response Is Less Effective in OB Subjects with Persisting Periodontal Inflammation

To explore periodontal treatment response according to BMI, we compared clinical parameters before and after standard periodontal treatment (i.e., oral hygiene instructions, scaling and root planning) in NW and OB subjects. We observed a significant decrease in periodontal probing depth (*p* < 0.0001 vs. *p* = 0.003), clinical attachment loss (*p* = 0.001 vs. *p* = 0.004), bleeding on probing (*p* = 0.0001 vs. *p* = 0.013) and number of periodontal pockets ≥ 5 mm/number of sites examined (*p* < 0.0001 vs. *p* = 0.001) in both groups (Table 2).

We also compared periodontal inflammation, measured by the PISA index, before and after periodontal treatment in each group. Normo-weighted subjects showed a highly significant decrease in periodontal inflammation (PISA index) (986.99 mm^2^ ± 591.47 vs. 477.28 mm^2^ ± 542.55, *p* = 0.0007) after periodontal treatment, shifting from a severe state of periodontitis to a mild state of periodontitis (Figure 3). Obese patients also showed a significant decrease in periodontal inflammation (PISA index) after periodontal treatment (1677.83 ± 1489.92 mm^2^ vs. 741.17 mm^2^ ± 1305.56, *p* = 0.03); however, the reduction in inflammation was lesser, shifting from a severe to a moderate state of periodontitis corresponding to a still unstable periodontal condition. Additionally, there seemed to be a broader variability in periodontal treatment response in OB subjects compared to NW subjects.

## 3. Discussion

In our study, we showed that obesity was associated with more severe periodontal inflammation according to Periodontal Inflamed Surface Area (PISA) index. In addition, obese subjects with periodontitis were characterized by a specific signature of periodontal microbiota with an increase in Gram-positive bacteria in periodontal pockets, associated with a decrease in microbiota diversity. Finally, periodontal treatment was less effective in OB subjects with persisting periodontal inflammation (PISA index), revealing a still unstable periodontal condition and at risk of recurrence.

The aim of this study was to analyze the subgingival microbiota status of obese subjects with periodontitis to understand the link between periodontitis and obesity. Many studies suggested an increase in Gram-negative in obese subjects with periodontitis using genomic DNA probes targeting a limited group of bacteria [28,29,30]. We showed that obese subjects with periodontitis were characterized by a specific signature of periodontal microbiota with a decrease in Gram-negative bacteria and an increase in Gram-positive bacteria in periodontal pockets compared to normo-weighted subjects. The relative abundance of periodontal pathogen bacteria such as Porphyromonadaceae and Tannerellaceae was significantly lower in periodontal microbiota of obese subjects than those of normo-weighted subjects. Our results contradict a recent study published in 2023 that used taxonomic analysis of the subgingival microbiota. They observed an enrichment of *Aggregatibacter actinomycetemcomitans Tannerella forsythia* and *Treponema denticola* in obese and overweight patients with moderate–severe periodontitis [31]. We wondered about the origin of this increase in Gram-positive bacteria in our obese periodontal specimens. Because of their proximity, we hypothesized that local inflammation could facilitate the passage of Gram-positive salivary bacteria from the salivary microbiota to the periodontal pockets and, conversely, the passage of bacteria from the periodontal pocket to the salivary microbiota. To this end, we compared salivary and periodontal microbiota in relation to obesity. In obese subjects with periodontitis, a similar proportion of Gram-positive bacteria was observed in the salivary and periodontal microbiota. Concerning the beta diversity, we observed that the composition of the periodontal microbiota of normo-weighted subjects was different from that of the salivary microbiota. In obese subjects, on the other hand, it appeared to share similarities with that of the salivary microbiota. The periodontal and salivary microbiota come from and evolve in two very distinct environments. The translocation of these Gram-positive bacteria in the periodontal microbiota from the saliva in obese subjects with periodontitis could be linked to the low-grade inflammatory environment characteristic of obese subjects.

Our data showed that obese subjects presented a more severe state of periodontal inflammation, assessed by the PISA index, compared to normo-weighted subjects, irrespective of their oral hygiene habits [32,33]. The PISA index was developed for use as a key parameter in periodontal medicine studies, quantifying the periodontal inflammatory load, which represents a major component of the biological hypothesis that may explain the potential association between periodontitis and systemic diseases [27,34]. It reflects the inflammatory surface of the pocket epithelium in square millimeters, based on linear clinical measurements such as clinical attachment loss and gingival recession, as well as on the assessment of periodontal disease activity, the bleeding on probing index. Our results were in line with previous studies concerning higher gingival inflammation in people with a higher BMI [32,33]. BMI does not take into account body fat distribution, which may be responsible for certain pathologies associated with obesity [35,36]. For example, abdominal obesity is strongly associated with cardiovascular and metabolic complications [37,38]. Other obesity assessment parameters such as waist circumference and waist-to-hip ratio have also been found to be positively associated with clinical parameters of gingival inflammation [39,40,41]. This increased local inflammation in obese people may be partly explained by the chronic low-grade systemic inflammatory state that characterizes obesity. Indeed, in obese individuals, adipocyte hypertrophy per se leads to abnormal adipocyte function and adipose tissue remodeling. The altered adipocyte secretion profile is defined by increased secretion of pro-inflammatory cytokines (TNF, IL-6, IL-1β) and decreased secretion of adiponectin [42,43]. These inflammatory cytokines from adipose tissue also lead to the recruitment of various immune cells such as macrophages and T lymphocytes, which also help to maintain adipose tissue inflammation. This immuno-inflammatory infiltration contributes to the maintenance of low-grade systemic inflammation [44]. The persistence of low-grade systemic inflammation can also have an impact locally, leading to an increase in pro-inflammatory mediators in periodontal tissues [45,46,47]. Indeed, higher levels of pro-inflammatory cytokines (IL-1β, IL-6, TNF) were found in blood and gingival fluid in obese subjects with periodontitis compared to normo-weighted subjects with periodontitis [17,18]. Higher levels of pro-inflammatory adipokines (leptin, visfatin and resistin) were also found in greater quantities in the gingival fluid of obese subjects [19]. Furthermore, the low-grade systemic inflammation associated with obesity not only increases periodontal inflammation but promotes bone resorption processes caused by the recruitment of immuno-inflammatory cells and certain cytokines (IL-1β, TNF) [20]. In addition, endotoxemia, a consequence of salivary and periodontal bacteria release, activates innate and adaptive immunity, triggering local and systemic inflammation through lipopolysaccharides (LPS), major components of Gram-negative bacterial outer membranes. LPS stimulate the production of inflammatory mediators, cytokines and matrix metalloproteinases, leading to periodontal tissue destruction. LPS translocation into the bloodstream causes endotoxemia and leads to the M1/M2 polarization of macrophages in adipose tissues [48]. In humans, expression of M1 macrophages in the adipose tissue correlates with body mass index (BMI) and, therefore, obesity, thereby promoting adipose tissue inflammation. Therefore, chronic low-grade inflammation and macrophage accumulation in adipose tissue induced by the dysbiosis of the salivary microbiota contribute to the aggravation of oral and systemic diseases such as periodontal disease and obesity [49]. Obesity can also impair the immune response to oral bacteria. Free fatty acids also bind to Toll-Like Receptor 4 (TLR-4), resulting in tolerance leading to an inappropriate immuno-inflammatory response to bacterial aggression and thus worsening periodontal destruction [50,51]. Periodontitis is a complication of systemic inflammation in obesity, just as diabetes is. Additionally, the inflamed periodontal pocket in obese and diabetic subjects can be compared to, in terms of systemic inflammation complication, foot ulcers in type 2 diabetes [52].

Our results also showed that non-surgical periodontal treatment improved oral clinical parameters in both obese and normo-weighted subjects. However, in obese subjects, periodontal inflammation persisted, thus increasing the risk of recurrence. Several studies have evaluated the impact of obesity on short-term periodontal treatment response, with conflicting conclusions [53]. Some have shown that obesity does not interfere with response to treatment, nor, in particular, with the reduction in clinical periodontal parameters such as pocket depth [54,55]. On the other hand, other studies have shown that the improvement in periodontal conditions after non-surgical periodontal therapy was less important in obese patients than in normo-weighted patients [53]. Suvan et al. evaluated the situation at 6 months after periodontal therapy and demonstrated that obese patients had a significantly lesser improvement in periodontal parameters than normo-weighted patients [56]. What is more, even when periodontal parameters improved, the values in obese patients remained higher than those in normo-weighted patients and above baseline [57]. Finally, a recent study showed that obesity is also associated with a higher risk of tooth loss over five years, underlining the fact that these patients, considered to be at risk, require closer monitoring [58,59]. In our study, obese patients had a significantly higher PISA index than normo-weighted patients at baseline. Non-surgical periodontal therapy significantly improved this parameter. However, post-therapy periodontal inflammation in obese patients remained at high levels, reflecting a still moderate, non-stabilized periodontitis condition [27]. The persistence of periodontal inflammation in the obese can be explained by the specific signature of the subgingival microbiota and the role of systemic inflammation. It could also be consistent with a persisting break of the epithelial barrier of the gingiva, suggesting that remaining periodontal inflammation in obese patients increases the risk of periodontitis recurrence [60]. Indeed, we observed a specific signature of the periodontal microbiota in obese patients with periodontitis, with an increase in the proportion of Gram-positive bacteria. Theoretically, the aim of non-surgical periodontal treatment is to disorganize the subgingival bacterial biofilm and reduce the proportion of periodontopathogen bacteria (classically Gram-negative and anaerobic bacteria). Mechanical removal of the subgingival biofilm leaves bacteria vulnerable to antibacterial agents (antiseptics) and to oxygen supply [61,62]. Because of the local periodontal pocket persisting inflamed conditions in obese subjects and their specific microbial composition, periodontal therapy might no longer be sufficient for optimal periodontal management in the obese. Adjuvant therapies such as the use of probiotics or prebiotics could be one solution to counterbalance the proportion of Gram-positive bacteria and to more easily recover eubiosis of the microbiota. Several strains of probiotic bacteria from the Bifidobacteriaceae and Lactobacillacea families have been found to be of interest for periodontal diseases. *Bifidobacterium* species have been shown to exert competitive exclusion of neighboring microbes due to their rapid proliferation. Additionally, *Lactobacillus* species have been found to enhance and modulate the host’s immune response, as well as to have an anti-microbial activity against periodontopathogens, particularly *Porphyromonas gingivalis*, *Fusobacterium nucleatum* and *Agregatibacter actinomycetemcomittans* [63,64]. In addition, increased inflammation, due to obesity, can modify the environment and have an impact on periodontal dysbiosis, as well as play a role in poorer wound healing. Once again, we can draw a parallel with diabetes and diabetic foot ulcers. A difference in microbiota composition between diabetic and non-diabetic foot ulcers has been demonstrated, along with a poorer response to treatment in diabetics [65,66,67,68]. This alteration of the skin microbiota at the level of pressure sores in diabetics is thought to be due to diabetes (alteration of host immunity and changes in the microenvironment caused by hyperglycemia) and implies the implementation of new possible prevention or therapy strategies for skin complications in these patients [65,66,67,68]. Similarly, the management of periodontal disease in obese patients requires a global approach. Even if our results are based on DNA sequencing, which is an indirect biomarker of microbial gene expression and should be interpreted with this limit, a better understanding of the dysbiosis of the periodontal microbiota in these patients, and the identification of the molecular players and metafactors of the oral microbiota, could provide a better understanding of the pathophysiological mechanisms associated with obesity. Finally, the management of host response modulation in the context of obesity and the management of systemic inflammation are also important fields of research if we are to improve the management of this population.

## 4. Materials and Methods

We followed the STROBE statement guidelines for reporting observational studies [69]. We conducted a cross-sectional study from September 2020 to February 2023 in the Periodontology department in Toulouse’s Public Teaching Hospital (CHU de Toulouse, France). All participants gave their informed consent, non-opposition for the study and agreement for an oral examination and microbiota sampling for biological analyses. This study was approved by the French Committee for the Protection of Individuals (CPP, ID-RCB: 2020-A03496-33).

### 4.1. Participants

We included adult patients with periodontitis. After explanation of the protocol, an informed consent form was signed by each patient of the cohort in order to allow data collection and analysis.

Non-inclusion criteria included liver disease or steatosis (with an etiology of viral infections and autoimmune disease), one or more known chronic viral infections (HIV, HBV, HCV and mononucleosis), chronic renal insufficiency (creatinine clearance less than 60 mL/min), chronic or acute gastrointestinal disease and a history of gastrointestinal surgery modifying the anatomy, as well as taking prebiotics, probiotics or antibiotics in the 3 months prior to inclusion. Finally, minors and people under guardianship or trusteeship, as well as pregnant or breastfeeding women, were not included in the study. We also excluded patients who presented diabetes (in order to purely explore obesity).

### 4.2. Variables and Data Collection

#### 4.2.1. General and Socio-Demographic Characteristics

General data were collected from a questionnaire filled out at the time of the first consultation. They concerned: the patient’s age, gender, height and weight, used to calculate the body mass index (BMI) (weight (kg)/height^2^ (m^2^)) and Socio-Professional Category (SPC) (classification according to INSEE’s nomenclature of professions and socio-professional categories; https://www.insee.fr/fr/information/6208292 (accessed on 30 August 2023).

The hygienic and dietary behavior was evaluated by using a questionnaire that focused on the patient’s lifestyle and oral habits. It was given to the patient on the day of inclusion. The data recorded were: stress evaluated using the VAS (Visual Analogy Scale) from 0 to 10, tobacco consumption, oral hygiene habits (visits to the dental surgeon and toothbrushing habits).

#### 4.2.2. Oral Health Characteristics

Two experienced periodontists (C.T., S.L.-D.) conducted the oral health examinations. They were calibrated for full-mouth periodontal charting and periodontal disease assessment (probing depth, periodontal attachment loss and BOP) before the onset of the study with an overall inter-rater kappa reaching 0.9. Prior to inclusion, all of the examiners were trained in a calibration process that focused on caries, periodontal status and periodontal treatment.

For each patient, complete oral and periodontal examination was performed. The oral status was evaluated by counting the number of decayed, missing and filled teeth, excluding wisdom teeth (maximum 28 teeth). The DMFT (Decayed–Missing–Filled Teeth) index, validated by the WHO (World Health Organization), was calculated [70]. The periodontal status was evaluated by taking into account different criteria.

For each patient, complete oral and periodontal examinations were carried out. In order to conduct full-mouth periodontal charting (6 measurements per tooth), a PCP15 probe was used. Different periodontal indices were then determined and calculated for each participant: probing depth (distance in millimeters from the gingival margin to the bottom of the periodontal pocket), clinical attachment loss (distance in millimeters from the cementoenamel junction of the tooth to the bottom of the periodontal pocket), bleeding on probing (BOP) index (percentage of teeth presenting gingival bleeding), plaque index (percentage of teeth with plaque) and the PISA (Periodontal Inflamed Surface Area) [34,71,72]. The PISA index reflects the surface area of inflamed tissue found in hemorrhagic pockets, in square millimeters, and takes into account the level of clinical attachment of loss, recessions and bleeding on probing (BOP) identified during the probing assessment.

These patients underwent standard periodontal treatment (oral hygiene instructions, scaling and root planning). Their periodontal status was reassessed between 8 and 12 weeks after non-surgical treatment for the full mouth. At this point, a full-mouth periodontal charting was performed, and the same parameters were measured.

#### 4.2.3. Oral Microbiota Analysis

On the day of the oral examination, unstimulated saliva was collected into sterile tubes. Periodontal microbiota was collected using sterile paper tips inserted into the periodontal space for 30 s. Salivary and periodontal microbiota were collected at the same time on the same day and then frozen into liquid nitrogen and stored at −80 °C until taxonomic analysis of the oral microbiota was performed [73,74]. Total DNA was extracted from frozen saliva using a Qiamp Cador Pathogen Mini kit (QIAGEN ref 54106, Les Ulis, France) according to the manufacturer’s recommendations. Then, hypervariable V3 to V4 regions of the 16S bacterial ribosomal DNA (16S rDNA) were analyzed as previously described [23,74]. The Miseq sequencing of the samples was performed by Vaiomer (Labège, France).

##### Bacterial 16S rRNA Gene Sequencing

Vaiomer (Labège, France) performed the profiling of the oral microbiome. The V3–V4 hypervariable regions of the 16S rRNA gene were amplified by PCR using universal Vaiomer primers [31]. Amplicons (467 bp on the Escherichia coli reference genome) were purified using the magnetic beads CleanNGS for DNA clean-up (CleanNA, Waddinxveen, The Netherlands). To generate an equivalent number of raw reads, all libraries were pooled in the same quantity and were sequenced on a MiSeq Illumina platform (2 × 300 bp paired-end MiSeq kit v3, Illumina, San Diego, CA, USA) [75].

##### 16rRNA Gene Sequence Analysis

A bioinformatic pipeline based on ‘find, rapidly, OTUs with Galaxy solution’ (FROGS) guidelines [32] was used to analyze the targeted metagenomic sequences. Single-read sequences were cleaned after demultiplexing of barcoded Illumina paired reads, and the last 10 and 50 bases of R1 and R2 read, respectively, were trimmed and paired into longer fragments. OTUs were produced with single-linkage clustering. The use of BLAST against the SILVA 138.1 database allowed taxonomic assignment to determine bacterial profiles from phylum to genus and species level, where reachable. The following filters were applied: amplicons with a length of < 350 nt or a length of > 500 nt were removed. OTUs with an abundance lower than 0.005% and that appeared less than twice in the entire dataset were removed. Alpha and beta diversity analyses were conducted on the OTU table. The data for this study have been deposited in the European Nucleotide Archive (ENA) at EMBL-EBI under accession number PRJEB66524 (https://www.ebi.ac.uk/ena/browser/view/PRJEB66524).

### 4.3. Statistical Analysis

Significant variations in alpha diversity were assessed using the Kruskal–Wallis test or the Wilcoxon rank-sum test. Multidimensional scaling analyses (MDS) were performed on beta diversity distance matrices, and differences between groups were assessed using PERMANOVA and PERMDISP analyses (2000 permutations). LEfSe (linear discriminant analysis effect size) analyses based on non-parametric tests were used to determine significant variations in taxa relative abundance [23].

The results were presented in the form of means and standard deviations for the quantitative variables and numbers, with proportions for the qualitative variables. The comparison between all the groups was ensured by the application of an ANOVA analysis (total *p*-value) after verification of the hypothesis of a normal distribution of the values and the equality of the variances. Otherwise, a non-parametric Kruskal–Wallis test was preferred. The comparison of the groups two by two was ensured by the use of Student’s *t*-tests after testing the same hypotheses. If not, a non-parametric Mann–Whitney Wilcoxon test was preferred. The qualitative comparison between the groups was carried out by means of Fisher’s exact test. The significance level was set at 5%. The database was created using Microsoft Excel^®^, and the analyses and figures were made using Stata v.13^®^, R v.3.5.1^®^ and GraphPad Prism 5^®^.

## 5. Conclusions

The imbalance of the inflammatory immune response is a common parameter between chronic metabolic diseases, of which obesity is one, and periodontal disease. It represents a hypothesis explaining their relationship. To our knowledge, this is the only study to have sequenced both salivary and subgingival microbiota in obese and normal-weighted subjects. Further studies are needed to explain the possible translocation and persistence of specific salivary Gram-positive bacteria into the periodontal pocket. Finally, if we consider systemic inflammation to play a central role in pathogenesis, periodontitis could merely be a local clinical manifestation of systemic inflammation, representing a symptom of existing chronic metabolic disorders [47].

## Figures and Tables

**Figure 1 ijms-24-15123-f001:**
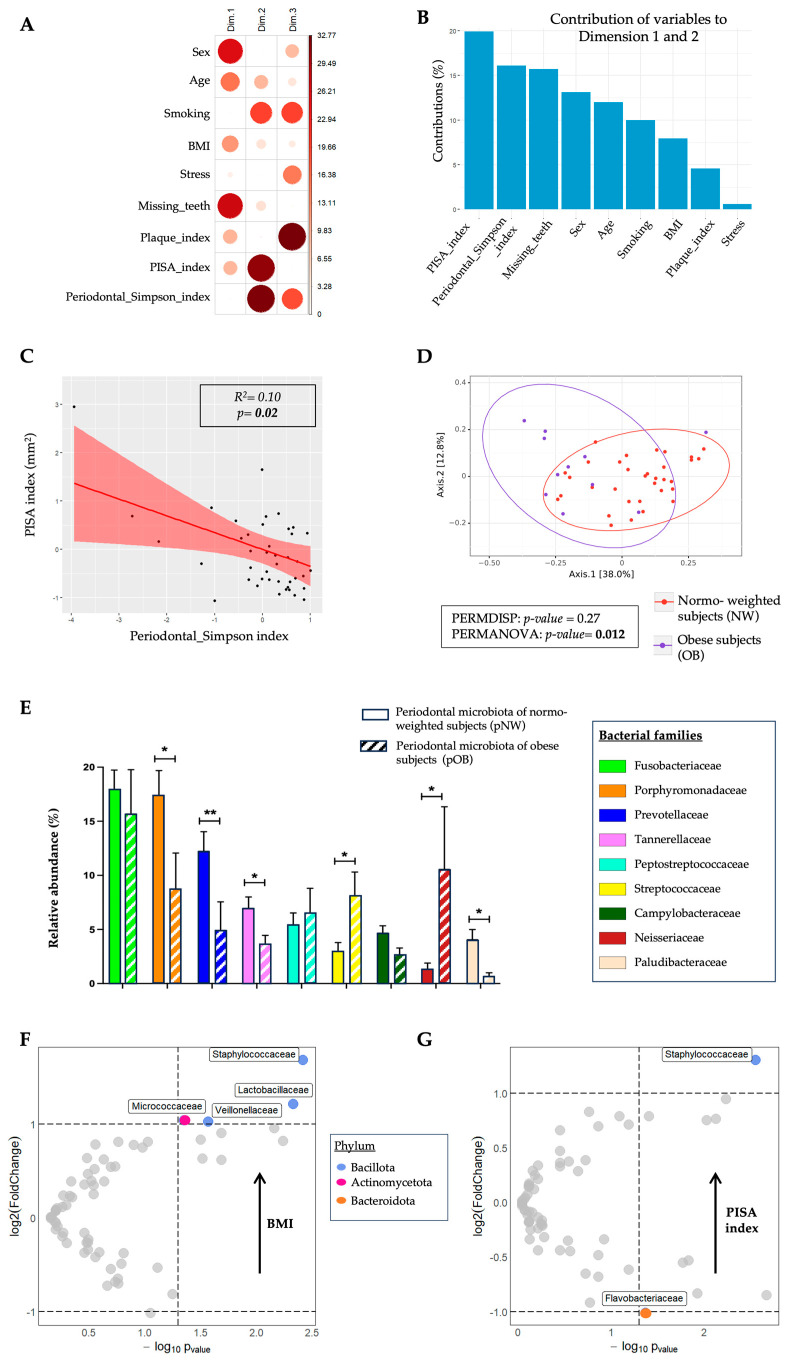
Comparison of periodontal microbiota between obese subjects (OB; *n* = 11) and normo-weighted subjects (NW; *n* = 34). (**A**,**B**) Principal Component Analysis (PCA) between oral clinical parameters and Periodontal Simpson Index (alpha diversity) with their contribution for each dimension. (**C**) Pearson’s correlation analysis between PISA index and Periodontal Simpson Index. (**D**) Unifrac index representation of the beta diversity. (**E**) Relative abundance (%) for taxonomic family, identified with significant differences in periodontal microbiota between obese and normo-weighted subjects. Data as mean ± SD, * *p* < 0.05, ** *p* < 0.01, unpaired Mann–Whitney test. (**F**) Volcano plot analysis identifying significant changes in bacterial abundance according to increasing BMI. (**G**) Volcano plot analysis identifying significant changes in bacterial abundance according to increasing PISA index.

**Figure 2 ijms-24-15123-f002:**
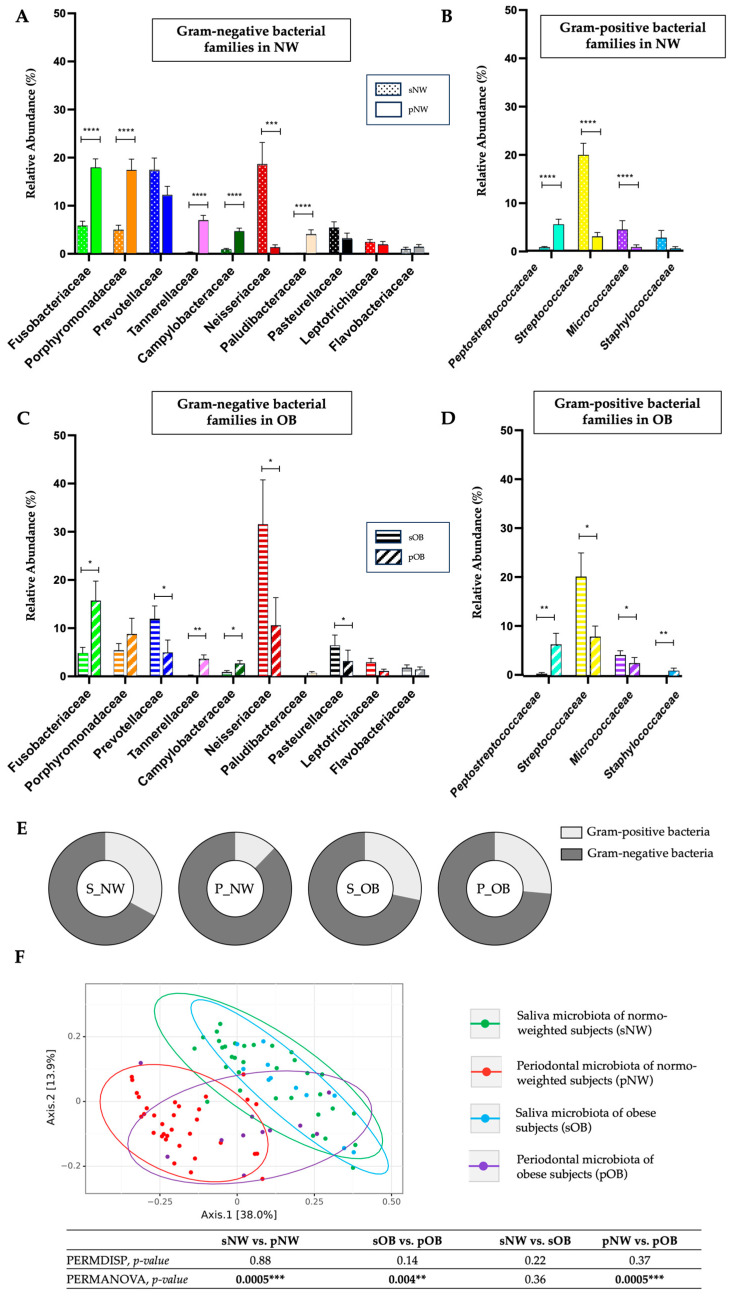
Comparison of periodontal and saliva microbiota in obese subjects (OB; *n* = 11) and normo-weighted subjects (NW; *n* = 34). (**A**,**B**). Relative abundance (%) for taxonomic family, identified with significant differences in periodontal and saliva microbiota in normo-weighted subjects (NW). (**C**,**D**) Relative abundance (%) for taxonomic family, identified with significant differences in periodontal and saliva microbiota in obese subjects (OB). (**E**) Proportion of Gram-positive and Gram-negative bacteria in each group. (**F**) Unifrac index representation of the beta diversity. Data as mean ± SD, * *p* < 0.05, ** *p* < 0.01, *** *p* < 0.001, **** *p* < 0.0001, unpaired Mann–Whitney test.

**Figure 3 ijms-24-15123-f003:**
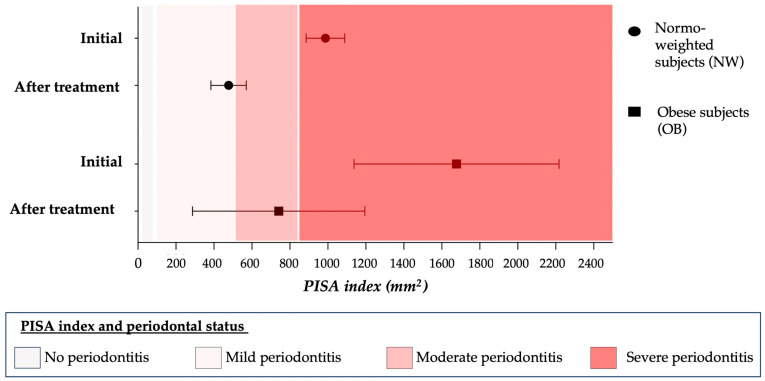
Comparison of PISA index before and after periodontal treatment between obese and normo-weighted subjects [27].

**Table 1 ijms-24-15123-t001:** General and oral parameters between normo-weighted subjects (NW; *n* = 34) and obese subjects (OB; *n* = 11). Data as mean ± SD, * *p* < 0.05, **** *p* < 0.0001, unpaired Mann–Whitney test and Fisher’s test exact. Significant *p*-values have been highlighted in bold.

Parameters	Normo-Weighted Subjects (NW)	Obese Subjects (OB)	*p*-Value
*n*	34	11	
Sex (number of women, %)	26 (76.47%)	6 (54.55%)	0.25
Age (years)	47.56 ± 10.62	53.64 ± 11.96	0.49
Height (cm)	166.88 ± 9.49	169.18 ± 9.73	>0.99
Weight (kg)	60.5 ± 9.86	91.91 ± 9.30	**<0.0001 ******
Smoking (number of smokers, %)	13 (38.24%)	3 (27.27%)	0.72
Stress on a scale from 0 to 10 (EVA)	5.03 ± 3.17	5.36 ± 2.98	0.76
DMF index	11.06 ± 5.35	11.00 ± 6.68	>0.99
Number of decayed teeth	0.32 ± 0.84	0.63 ± 1.50	0.39
Number of missing teeth	2.38 ± 2.94	4.00 ± 4.58	0.18
Number of filled teeth	8.32 ± 4.40	6.36 ± 5.08	0.22
Probing depth (mm)	3.74 ± 0.88	3.72 ± 1.16	0.95
Loss of attachment (mm)	4.64 ± 1.24	5.06 ± 2.05	0.40
Plaque index (%)	12.12 ± 14.82	16.09 ± 13.87	0.44
Bleeding on probing (%)	28.71 ± 15.05	33.91 ± 21.14	0.37
Number of periodontal pockets ≥ 5 mm/numbers of sites studied	0.31 ± 0.18	0.29 ± 0.24	0.82
PISA index (mm^2^)	986.99 ± 591.47	1677.83 ± 1489.92	**0.049 ***
Brushing frequency			0.42
Once a day	7 (20.59%)	2 (5.88%)	
Two times a day	2 (7.69%)	4 (15.38%)	
Three times a day	4 (36.36%)	1 (9.09%)	
Dental check-up frequency			0.71
Two times a year	11 (33.33%)	4 (36.36%)	
Once a year	12 (36.36%)	4 (36.36%)	
Once every 2 years	7 (21.21%)	1 (0.09%)	
Less than once every 2 years	3 (9.09%)	2 (18.18%)	

**Table 2 ijms-24-15123-t002:** Oral parameters before and after periodontal treatment in normo-weighted subjects (NW; *n* = 34) and obese subjects (OB; *n* = 11). Data as mean ± SD, * *p* < 0.05, ** *p* < 0.01, *** *p* < 0.001 **** *p* < 0.0001, unpaired Mann–Whitney test and Fisher’s exact test. Significant *p*-values have been high-lighted in bold.

Parameters	Initial	After Periodontal Treatment	*p*-Value
Normo-weighted subjects (*n* = 34)			
Probing depth (mm)	3.74 ± 0.88	2.91 ± 0.59	**<0.0001 ******
Loss of attachment (mm)	4.64 ± 1.24	4.07 ± 1.04	**0.001 ****
Plaque index (%)	12.12 ± 14.82	12.65 ± 15.92	0.92
Bleeding on probing (%)	28.71 ± 15.05	12.24 ± 10.04	**0.0001 *****
Number of periodontal pockets ≥ 5 mm/numbers of sites studied	0.31 ± 0.18	0.13 ± 0.11	**<0.0001 ******
Obese subjects (*n* = 11)			
Probing depth (mm)	3.72 ± 1.16	2.80 ± 0.90	**0.003 ****
Loss of attachment (mm)	5.06 ± 2.05	4.22 ± 1.82	**0.004 ****
Plaque index (%)	16.09 ± 13.87	21.09 ± 16.94	**0.11**
Bleeding on probing (%)	33.91 ± 21.14	16.27 ± 19.15	**0.013 ***
Number of periodontal pockets ≥ 5 mm/numbers of sites studied	0.29 ± 0.24	0.11 ± 0.20	**0.001 ****

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
