# Peer review of "Obesity Is Associated with the Severity of Periodontal Inflammation Due to a Specific Signature of Subgingival Microbiota"

_ijms, 2023, doi:10.3390/ijms242015123_

Round 1
Reviewer 1 Report
Title: “Obesity is associated with the severity of periodontal inflammation due to a specific signature of subgingival microbiota”
In this study, the authors showed that periodontal microbiota taxonomic analysis showed that the obese subjects with periodontitis were characterized by a specific signature of subgingival microbiota with an increase of Gram-positive bacteria in periodontal pockets, associated with a decrease of microbiota diversity compared to normo-weighted subjects with periodontitis. Finally, periodontal treatment response was less effective in OB subjects with persisting periodontal inflammation reflecting a still unstable periodontal condition and a risk of recurrence.
Minor Points:
1: TNF-α is only TNF
2: Once the main aim of this study was to analyse the subgingival microbiota of obese subjects, and after authors said “Unstimulated saliva was collected into sterile tubes on the day of the oral examination. Periodontal microbiota was collected using sterile paper tips inserted into the periodontal space for 30 seconds”, so saliva and periodontal microbiota should be collected at the same day and it is not clear on MM.
3. The section numbering in the Materials and Methods appears to be unclear, and I recommend a more structured approach by aligning it with the section numbers (e.g., 4.1, 4.1.1, 4.2, 4.2.1, etc.).
4. I suggest authors to use the new bacteria phyla names - NCBI Taxonomy (as seen in Fig. 1F):
Firmicutes: Bacillota
Actinobacteria: Actinomycetota
Bacteroidetes: Bacteroidota
Major points:
1: The salivary microbiota, composed of bacteria shed from oral surfaces, is a complex and individualized community that exhibits temporal stability while being significantly influenced by dietary and lifestyle factors. It reflects local bacterial changes in both supragingival and subgingival microbiota. However, the hypothesis that Gram-positive bacteria in saliva directly contribute to Gram-positive bacteria in obese samples should be critically reassessed. The salivary microbiota is primarily shaped by the supragingival biofilm rather than the subgingival one, emphasizing the need to consider this distinction when discussing the potential translocation of salivary microorganisms to the subgingival environment. While it's theoretically plausible that Gram-positive salivary bacteria could migrate to periodontal pockets due to local inflammation, it's equally important to acknowledge the possibility of the reverse scenario where subgingival bacteria colonize the oral cavity and impact the salivary microbiota, aligning better with established microbiological principles and observations.
2. In light of the findings presented in the manuscript, there are some important considerations regarding the observed changes in the relative abundance of Gram-positive bacterial families in relation to the PISA index. The manuscript reports a significant increase in the relative abundance of the Staphylococcaceae family with a concomitant rise in the PISA index, while the relative abundance of the Flavobacteriaceae family appears to increase as the PISA index decreases.
However, upon closer examination of Figure 2B and 2D, it is evident that Staphylococcaceae did not display a substantial abundance in the samples, which raises questions about its potential role in the context of the observed trends. Additionally, it is noteworthy that the Flavobacteriaceae family primarily comprises environmental bacteria. This raises an inconsistency that warrants further discussion.
3. The use of DNA for microbiome analysis in the study assumes the inclusion of all bacterial types, whether active, inactive, or dead, making it challenging to pinpoint the specific role of Gram-positive bacteria in the context of periodontal inflammation. Furthermore, the concept that the abundance of Gram-positive bacteria alone plays a pivotal role in this particular microbiome dysbiosis may oversimplify the complex microbial interactions at play in the periodontal environment. To gain deeper insights into the functional contributions of these Gram-positive bacteria, a transcriptome analysis may be instrumental, allowing us to assess their metabolic activities and gene expression patterns, thereby shedding light on their potential role as key players or bystanders in periodontal inflammation. This approach would offer a more nuanced understanding of the microbiome dynamics at the site and strengthen the study's overall conclusions.
4. The authors' exploration of adjuvant therapies, including probiotics, to counterbalance the proportion of Gram-positive bacteria in the manuscript is indeed intriguing. It's worth noting that numerous bacteria, particularly Gram-positive strains like Lactobacillus and Streptococcus, have demonstrated probiotic effects in promoting oral health. However, the mechanism underlying the counterbalance of Gram-positive bacteria through probiotics requires further elucidation. An in-depth discussion of the specific strains, their potential interactions within the oral microbiome, and their mechanisms of action, such as competitive exclusion or modulation of local immune responses, would enhance the manuscript's scientific rigor. Addressing this aspect more comprehensively will contribute to a more insightful exploration of probiotics as a potential therapeutic strategy in the context of Gram-positive bacterial imbalances in oral health.
Author Response
Dear Editor, Dear Reviewers
Thank you for giving us the opportunity to send back a revised version of our manuscript entitled “Obesity is associated with the severity of periodontal inflammation due to a specific signature of subgingival microbiota.” ijms-2635015
We have carefully studied the comments made by the two reviewers and have modified the text accordingly. We also included additional data and modified sentences as suggested in red color in the text.
Thus, we hope that you will now find our revised version suitable for publication in your journal.
Yours sincerely,
Charlotte Thomas and Vincent Blasco-Baque; DMD and PhD
Response to Reviewer 1
The Authors thank the reviewer for all the insightful comments and his/her careful review.
Minor Points:
Point 1 : TNF-α is only TNF
ïƒ We have corrected the text
Point 2: Once the main aim of this study was to analyse the subgingival microbiota of obese subjects, and after authors said “Unstimulated saliva was collected into sterile tubes on the day of the oral examination. Periodontal microbiota was collected using sterile paper tips inserted into the periodontal space for 30 seconds”, so saliva and periodontal microbiota should be collected at the same day and it is not clear on MM.
ïƒ We thank the reviewer for this suggestion. Accordingly, we have modified the methods section and added the following sentence (line 482):
“Salivary and periodontal microbiota were collected at the same time on the same day”
Point 3. The section numbering in the Materials and Methods appears to be unclear, and I recommend a more structured approach by aligning it with the section numbers (e.g., 4.1, 4.1.1, 4.2, 4.2.1, etc.).
ïƒ We have modified the materials and methods sections following the suggestion
Point 4. I suggest authors to use the new bacteria phyla names - NCBI Taxonomy (as seen in Fig. 1F):
Firmicutes: Bacillota
Actinobacteria: Actinomycetota
Bacteroidetes: Bacteroidota
ïƒ We have modified the figure following the remark and we imputed the information in Figure 1.
Major points:
Point 1 : The salivary microbiota, composed of bacteria shed from oral surfaces, is a complex and individualized community that exhibits temporal stability while being significantly influenced by dietary and lifestyle factors. It reflects local bacterial changes in both supragingival and subgingival microbiota. However, the hypothesis that Gram-positive bacteria in saliva directly contribute to Gram-positive bacteria in obese samples should be critically reassessed. The salivary microbiota is primarily shaped by the supragingival biofilm rather than the subgingival one, emphasizing the need to consider this distinction when discussing the potential translocation of salivary microorganisms to the subgingival environment. While it's theoretically plausible that Gram-positive salivary bacteria could migrate to periodontal pockets due to local inflammation, it's equally important to acknowledge the possibility of the reverse scenario where subgingival bacteria colonize the oral cavity and impact the salivary microbiota, aligning better with established microbiological principles and observations.
ïƒ We thank the reviewer for his comment. We modified the text (lines 291) accordingly:
We wondered about the origin of this increase in gram-positive bacteria in our obese periodontal specimens.” Because of their proximity, we hypothesized that local inflammation could facilitate the passage of gram-positive salivary bacteria from the salivary microbiota to the periodontal pockets, and conversely, the passage of bacteria from the periodontal pocket to the salivary microbiota.” To this end, we compared salivary and periodontal microbiota in relation to obesity. In obese subjects with periodontitis, a similar proportion of gram-positive bacteria was observed in the salivary and periodontal microbiota. Concerning the beta diversity, we observed that the composition of the periodontal microbiota of normo-weighted subjects was different from the salivary microbiota. In obese subjects, on the other hand, it appeared to share similarities with that of the salivary microbiota. The periodontal and salivary microbiota come from and evolve in two very distinct environments. The translocation of these gram-positive bacteria in the periodontal microbiota from the saliva in obese subjects with periodontitis could be linked to the low-grade inflammatory environment characteristic of obese subjects.
Point 2. In light of the findings presented in the manuscript, there are some important considerations regarding the observed changes in the relative abundance of Gram-positive bacterial families in relation to the PISA index. The manuscript reports a significant increase in the relative abundance of the Staphylococcaceae family with a concomitant rise in the PISA index, while the relative abundance of the Flavobacteriaceae family appears to increase as the PISA index decreases.
However, upon closer examination of Figure 2B and 2D, it is evident that Staphylococcaceae did not display a substantial abundance in the samples, which raises questions about its potential role in the context of the observed trends. Additionally, it is noteworthy that the Flavobacteriaceae family primarily comprises environmental bacteria. This raises an inconsistency that warrants further discussion.
ïƒ We thank the reviewer for this smart remark. We analyzed the relative abundance Staphylococcaceae and Flavobacteriaceae in each group and in each microbiota. The Fig1F and G represents a Volcanoplot analysis presented significant bacterial abundance following the BMI and PISA Index. To explain the fold change in abundance, we added supplementary Table 1 and Table 2 showing the relative abundance of Staphylococcaceae and Flavobacteriaceae in periodontal and salivary microbiota. The relative abundance of Staphylococcaceae is significantly higher in periodontal microbiota of OB compared to periodontal microbiota of NW subjects (pOB 1,13 % ± 1,69 vs pNW 0,48 % ± 1,99 ; p=0.006).
We added a sentence in the results section (line 213).
“The relative abundance of Staphylococcaceae is significantly higher in periodontal microbiota of OB compared to periodontal microbiota of NW subjects (respectively 1,13 % ± 1,69 vs. 0,48 % ± 1,99; p=0.006) (Supp.Table1-2). The relative abundance of Staphylococcaceae is around 4.4-fold less important in periodontal microbiota than in salivary microbiota in normo-weight subject. In contrast, no change was observed between both microbiota in obese subjects.”
We added the asked data by the reviewer in Supplementary Table 1 and 2:
|
|
Relative Abundance of Staphylococcaceae |
p value |
||||
|
|
Periodontal microbiota |
Salivary microbiota |
|
|||
|
|
pNW (n=34) |
pOB (n=11) |
sNW (n=34) |
sOB (n=11) |
|
|
|
|
0 |
5,54917240202491 |
0 |
0 |
|
|
|
|
0 |
2,20167958748987 |
0 |
0,04770739
|
|
|
|
|
0,0017690974065032 |
1,30684174153421 |
0 |
0 |
|
|
|
|
0,0059349529160402 |
0,0507283136459164 |
42,4307036
|
0 |
|
|
|
|
0,00132329394328362 |
0,178422205659823 |
0 |
0 |
|
|
|
|
0 |
2,12809286223399 |
0 |
0 |
|
|
|
|
1,68074563988388 |
0,953972020984262 |
0 |
0 |
|
|
|
|
0,000772105376941845 |
0,113687659873272 |
0 |
0 |
|
|
|
|
1,7507311405793 |
0 |
11,1088317
|
0 |
|
|
|
|
0,625988490317779 |
0 |
20,4362078
|
0,00525127
|
|
|
|
|
0 |
0 |
0 |
|
|
|
|
|
0,003180358108323 |
|
0 |
|
|
|
|
|
0,00135987815491732 |
|
0 |
|
|
|
|
|
0 |
|
0 |
|
|
|
|
|
0 |
|
0 |
|
|
|
|
|
0 |
|
0 |
|
|
|
|
|
0 |
|
0 |
|
|
|
|
|
0,0873871249635887 |
|
0 |
|
|
|
|
|
0 |
|
0 |
|
|
|
|
|
11,5166551030563 |
|
0 |
|
|
|
|
|
0,792546967790361 |
|
0 |
|
|
|
|
|
0 |
|
0 |
|
|
|
|
|
0 |
|
0 |
|
|
|
|
|
0 |
|
20,4362078 0 |
|
|
|
|
|
0,00177604120415594 |
|
0 |
|
|
|
|
|
0,0039714058776807 |
|
0 |
|
|
|
|
|
0 |
|
0 |
|
|
|
|
|
0 |
|
0 |
|
|
|
|
|
0 |
|
0 |
|
|
|
|
|
0 |
|
0,00549843
|
|
|
|
|
|
0 |
|
0 |
|
|
|
|
|
0 |
|
15,423589
|
|
|
|
|
|
0 |
|
0 |
|
|
|
|
|
0 |
|
0 |
|
|
|
|
MEAN ± SD |
0,484533576 ± 1,996730174 |
1,134781527 ± 1,693212834 |
2,629695849 ± 8,428264644 |
0,004814424 ± 0,014312952 |
pNW vs pOB * p=0,006
|
|
|
RATIO Perio/Saliva |
-4,427272695 |
0,9957574 |
|
|
|
|
Supp.Table 1. Relative abundance of Staphylococcaceae in periodontal microbiota.
|
|
Relative Abundance of Flavobacteriaceae |
p value |
|||
|
|
Periodontal microbiota |
Salivary microbiota |
|
||
|
|
pNW (n=34) |
pOB (n=11) |
sNW (n=34) |
sOB (n=11) |
|
|
|
0,000777157778572206 |
0 |
0,260704692684468
|
1,45 |
|
|
|
0,308523562887653 |
0 |
0,730273141122914 |
0,0106016432547045 |
|
|
|
12,5004422743516 |
2,28956461644782 |
0,246347751360642 |
1,09434895511314 |
|
|
|
1,29381973569676 |
0,0710196391042829 |
0,0147048011175649
|
0,924304339365635 |
|
|
|
0,919689290582117 |
0,185197732457032 |
0,212862026704509 |
0,247912317327766 |
|
|
|
0,087830584561335 |
3,50269829956216 |
0,281728853157425 |
0 |
|
|
|
0,178988496714906 |
0,0780664501623782 |
0,207845044885685 |
6,58859470468432 |
|
|
|
6,56289570400568 |
2,12328423586846 |
0,176645593157307 |
3,20090572251955 |
|
|
|
0,0480749969953127 |
5,36531813510231 |
0 |
1,11051502145923 |
|
|
|
0 |
0,425408996612484 |
0 |
3,05624113847608 |
|
|
|
1,23046711790758 |
1,61754855994642 |
1,2126627521062 |
0,141168878312427 |
|
|
|
7,69964698024998 |
|
0,578679125079635 |
|
|
|
|
0,0720735422106179 |
|
1,01419878296146 |
|
|
|
|
0,152296885684093 |
|
0,028169014084507 |
|
|
|
|
0 |
|
0,313686643552914 |
|
|
|
|
0,36246476037052 |
|
2,24211629970159 |
|
|
|
|
2,39599276471407 |
|
3,19783573806881 |
|
|
|
|
0,0606855034469366 |
|
0 |
|
|
|
|
1,56167045648058 |
|
0,495635449030922 |
|
|
|
|
0 |
|
0,8 |
|
|
|
|
0 |
|
0 |
|
|
|
|
0,20995597697257 |
|
0,978574371652246 |
|
|
|
|
1,41497045305319 |
|
0,342625759227535 |
|
|
|
|
0,234279823821573 |
|
0,420066631258751 |
|
|
|
|
0,43690613622236 |
|
12,1785894347656 |
|
|
|
|
0,631453534551231 |
|
0,0558174150334904 |
|
|
|
|
0,062439295129735 |
|
1,34044393196887 |
|
|
|
|
5,31077024756974 |
|
0,257450460290217 |
|
|
|
|
0,301309905166672 |
|
0,789714450811383 |
|
|
|
|
0,0753078763178878 |
|
0,197943586077968
|
|
|
|
|
0,180813318777293 |
|
2,01454659620114 |
|
|
|
|
0,607889442840899 |
|
1,37857900318134
|
|
|
|
|
0,682790391576176 |
|
0,53194210014833 |
|
|
|
|
2,84802739086452 |
|
1,26008778139601 |
|
|
|
MEAN ± SD |
1,467674352 ± 2,73873931 |
1,423464242 ± 1,770878136 |
0,992955213 ± 2,105828083 |
1,62041752 ± 1,989519754 |
pNW vs pOB p=0,8307
|
|
RATIO Perio/Saliva |
0,323449911 |
-0,138361943 |
|
|
|
Supp.Table 2. Relative abundance of Flavobacteriaceae in periodontal microbiota.
Point 3. The use of DNA for microbiome analysis in the study assumes the inclusion of all bacterial types, whether active, inactive, or dead, making it challenging to pinpoint the specific role of Gram-positive bacteria in the context of periodontal inflammation. Furthermore, the concept that the abundance of Gram-positive bacteria alone plays a pivotal role in this particular microbiome dysbiosis may oversimplify the complex microbial interactions at play in the periodontal environment. To gain deeper insights into the functional contributions of these Gram-positive bacteria, a transcriptome analysis may be instrumental, allowing us to assess their metabolic activities and gene expression patterns, thereby shedding light on their potential role as key players or bystanders in periodontal inflammation. This approach would offer a more nuanced understanding of the microbiome dynamics at the site and strengthen the study's overall conclusions.
ïƒ We thank the reviewer for this smart remark. This is a wise idea to analyze gene expression rather than the genome itself. However, the literature regarding meta transcriptomics is still debating about the quality of the RNA sequencing. We have notified, for the sake of clarity and to notify readers, this limit in the discussion section page 406. “Even if our results are based on DNA sequencing which is an indirect biomarker of microbial gene expression and should be interpreted with this limit”
Point 4. The authors' exploration of adjuvant therapies, including probiotics, to counterbalance the proportion of Gram-positive bacteria in the manuscript is indeed intriguing. It's worth noting that numerous bacteria, particularly Gram-positive strains like Lactobacillus and Streptococcus, have demonstrated probiotic effects in promoting oral health. However, the mechanism underlying the counterbalance of Gram-positive bacteria through probiotics requires further elucidation. An in-depth discussion of the specific strains, their potential interactions within the oral microbiome, and their mechanisms of action, such as competitive exclusion or modulation of local immune responses, would enhance the manuscript's scientific rigor. Addressing this aspect more comprehensively will contribute to a more insightful exploration of probiotics as a potential therapeutic strategy in the context of Gram-positive bacterial imbalances in oral health.
ïƒ We agree with this reviewer. This is indeed an important debate. We will add a small chapter to the discussion section to mention this hypothesis in line 388. However, we will carefully make sure that we are not overstating this point to avoid major extrapolation or our data.
« Adjuvant therapies such as the use of probiotics or prebiotics could be one solution to counterbalance the proportion of gram-positive bacteria and to more easily recover eubiosis of the microbiota. Several strains of probiotic bacteria from the Bifidobacteriaceae and Lactobacillacea families have been found of interest as adjunctive therapies for periodontal diseases. Bifidobacterium species have been shown to exert competitive exclusion of neighboring microbes due to their rapid proliferation. Also, Lactobacillus species have been found to enhance and modulate the host’s immune response as well as having an anti-microbial activity against periodontopathogens particularly Porphyromonas gingivalis, Fusobacterium nucleatum and Agregatibacter actinomycetemcomittans. (Ref: DOI: 10.1111/prd.12343 and DOI: 10.1111/jcpe.12545”

Reviewer 2 Report
1. Obesity induces macrophage accumulation in adipose tissue, promotes chronic low-grade inflammation, and increases adipokines derived from adipocytes.---- identified Cd137 and Tmem26 to be the best markers for adipocyte identification.
2. How salivary microbiome affect the susceptibility of people with obesity to develop oral diseases and other systemic inflammatory disorders?
3. What is the role of the microbiome in energy regulation and metabolism?
4. Role of dysbiosis of salivary microbiota in inflammatory oral diseases?
Author Response
Dear Editor, Dear Reviewers
Thank you for giving us the opportunity to send back a revised version of our manuscript entitled “Obesity is associated with the severity of periodontal inflammation due to a specific signature of subgingival microbiota.” ijms-2635015
We have carefully studied the comments made by the two reviewers and have modified the text accordingly. We also included additional data and modified sentences as suggested in red color in the text.
Thus, we hope that you will now find our revised version suitable for publication in your journal.
Yours sincerely,
Charlotte Thomas and Vincent Blasco-Baque; DMD and PhD
Response to Reviewer 2
The Authors thank the reviewer for his/her comments and his/her questions which improved the discussion of the manuscript.
Point 1. Obesity induces macrophage accumulation in adipose tissue, promotes chronic low-grade inflammation, and increases adipokines derived from adipocytes.---- identified Cd137 and Tmem26 to be the best markers for adipocyte identification.
Point 2. How salivary microbiome affect the susceptibility of people with obesity to develop oral diseases and other systemic inflammatory disorders?
Point 4. Role of dysbiosis of salivary microbiota in inflammatory oral diseases?
ïƒ Thank you for these remarks, we have added a paragraph in the discussion (line 338) that can answer points 1, 2 and 4:
Endotoxemia, a consequence of salivary and periodontal bacteria release, activates innate and adaptive immunity, triggering local and systemic inflammation through lipopolysaccharides (LPS), major components of Gram-negative bacterial outer membranes. LPS stimulates the production of inflammatory mediators, cytokines, and matrix metalloproteinases, leading to periodontal tissue destruction. LPS translocation into the bloodstream causes endotoxemia and leads to the M1/M2 polarization of macrophages in adipose tissues (DOI: 10.1038/ni774). In humans, expression of M1 macrophages in the adipose tissue correlates with body mass index (BMI) and therefore obesity, thereby promoting adipose tissue inflammation. Therefore, chronic low-grade inflammation and macrophage accumulation in adipose tissue induced by the dysbiosis of the salivary microbiota and contributes to the aggravation of oral and systemic diseases such as periodontal disease and obesity (DOI: 10.2337/db09-0287).
Point 3. What is the role of the microbiome in energy regulation and metabolism?
ïƒ This is an important comment. We were the first to publish a molecular mechanism through which microbiota controls energy metabolism i.e. the role of LPS. Furthermore, major publications have been released demonstrating in humans the importance of microbiota on metabolic disease. We added some more information in the introduction section (ligne 79)
This endotoxemia, defined as an elevation of plasma levels of lipopolysaccharides (LPS), major components of Gram-negative bacterial outer membranes, combined with a chronic immuno-inflammatory response, may contribute to the development and/or aggravation of metabolic disorders[4]. The molecular mechanism through which microbiota controls energy metabolism is linked to the production of LPS in bloodstream. Our previous results showed a link between food intake and increased endotoxemia (ref: doi: 10.1093/ajcn/87.5.1219.; doi: 10.2337/db06-1491, doi: 10.1136/gutjnl-2015-309897). The metabolic endotoxemia by infusion of LPS targeted the adipose tissue by an increase of LPS and pro-inflammatory inside adipose cells. This modification of local environment of adipose tissue promoted an infiltration of macrophages (F4/80-positive cells) and upgraded the stockage of lipids inside adipose cells (Zelkha SA, Freilich RW, Amar S. Periodontal innate immune mechanisms relevant to atherosclerosis and obesity. Periodontol 2000 2010;54:207–21). Saito et al. proposed the bidirectional link between obesity, dysbiotic microbiota with Gram-negatives bacteria and periodontitis (Saito T, Shimazaki Y. Metabolic disorders related to obesity and periodontal disease. Periodontol 2000 2007;43:254–66.).
